# PEG-Delivered CRISPR-Cas9 Ribonucleoproteins System for Gene-Editing Screening of Maize Protoplasts

**DOI:** 10.3390/genes11091029

**Published:** 2020-09-02

**Authors:** Rodrigo Ribeiro Arnt Sant’Ana, Clarissa Alves Caprestano, Rubens Onofre Nodari, Sarah Zanon Agapito-Tenfen

**Affiliations:** 1CropScience Department, Federal University of Santa Catarina, Florianópolis 88034000, Brazil; rodrigoarnt@hotmail.com (R.R.A.S.); clarissacapre@gmail.com (C.A.C.); rubens.nodari@ufsc.br (R.O.N.); 2GenØk—Centre for Biosafety, Siva innovasjonssenter Postboks 6418, 9294 Tromsø, Norway

**Keywords:** gene editing, mutagenesis, genetically modified, genetically modified organism (GMO), crop breeding, ribonucleoprotein complex (RNP), genetic screening

## Abstract

Clustered regularly interspaced short palindromic repeats (CRISPR)-Cas9 technology allows the modification of DNA sequences in vivo at the location of interest. Although CRISPR-Cas9 can produce genomic changes that do not require DNA vector carriers, the use of transgenesis for the stable integration of DNA coding for gene-editing tools into plant genomes is still the most used approach. However, it can generate unintended transgenic integrations, while Cas9 prolonged-expression can increase cleavage at off-target sites. In addition, the selection of genetically modified cells from millions of treated ones, especially plant cells, is still challenging. In a protoplast system, previous studies claimed that such pitfalls would be averted by delivering pre-assembled ribonucleoprotein complexes (RNPs) composed of purified recombinant Cas9 enzyme and in vitro transcribed guide RNA (gRNA) molecules. We, therefore, aimed to develop the first DNA-free protocol for gene-editing in maize and introduced RNPs into their protoplasts with polyethylene glycol (PEG) 4000. We performed an effective transformation of maize protoplasts using different gRNAs sequences targeting the inositol phosphate kinase gene, and by applying two different exposure times to RNPs. Using a low-cost Sanger sequencing protocol, we observed an efficiency rate of 0.85 up to 5.85%, which is equivalent to DNA-free protocols used in other plant species. A positive correlation was displayed between the exposure time and mutation frequency. The mutation frequency was gRNA sequence- and exposure time-dependent. In the present study, we demonstrated that the suitability of RNP transfection was proven as an effective screening platform for gene-editing in maize. This efficient and relatively easy assay method for the selection of gRNA suitable for the editing of the gene of interest will be highly useful for genome editing in maize, since the genome size and GC-content are large and high in the maize genome, respectively. Nevertheless, the large amplitude of mutations at the target site require scrutiny when checking mutations at off-target sites and potential safety concerns.

## 1. Introduction

The development of technologies related to genetic improvement, such as transgenesis and more recently, genome editing, have changed the way humans have grown food for thousands of years. Today, the most promising tool to DNA manipulation is CRISPR (Clustered Regularly Interspaced Short Palindromic Repeats), a gene-editing technology that has been adapted from the bacterial immune system against viral infections [1]. Currently, a target site in the host genome can be reached by an endonuclease enzyme (i.e., Cas9) led by a guide RNA molecule (gRNA) that contains the target-specific sequence. This protein complex forms the CRISPR-Cas9 ribonucleoproteins (RNPs) gene-editing system. Cas9 introduces a site-specific double-stranded DNA break (DSB) followed by the natural cell repair of disrupted genome integrity by error-prone non-homologous end-joining (NHEJ) or homology-directed repair (HDR) [2]. Therefore, this tool allows the in vivo modification of the DNA at the gene sequence of interest, with unprecedented speed, and has made it a milestone in manipulating and producing living modified organisms.

Although much is already known about the principles of CRISPR-Cas9 genome editing, the likelihood of different outcomes in terms of resolution, efficiency, accuracy, and DNA modification structure has shown to be species-dependent. Various factors, including target site choice, gRNA design, the properties of the endonuclease, the type of DSB introduced, whether or not the DSB is unique, the quantity of the endonuclease and gRNA, and the intrinsic differences in DNA repair pathways in different species, tissues, and cells will result in differences in the mutation signatures generated in plant species [3].

In the case of maize, most researchers acknowledge that *Zea mays* L. is an ancient amphidiploid species with a duplicated chromosome number of *n* = 10 [4]. Multiple independent domestication events have also contributed to the genetic variability encountered in modern maize [4]. In addition, the genetic variants and the underlying mechanisms influencing variance heterogeneity in maize have so far hidden additive genetic effects and epistatic interaction effects in elite varieties [5]. Therefore, CRISPR can be a useful tool to access genomic regions in the maize genome, which are difficult to achieve by conventional breeding.

While CRISPR technology has already been tested on commercial crops to increase yield, drought tolerance and growth under limited nutrient conditions, improve nutritional properties and develop resistance to plant pathogens [6]; breeding and research of major monocotyledon species, more specifically maize, are still at its infancy. Maize has shown to be an exemption in the plant portfolio for the in silico analysis of potential Cas9 target sites as only 29.5% of annotated transcripts matched a specific gRNA [7]. Among eight analyzed plant species, maize had the largest genome, the highest GC content, and the greatest number of annotated transcripts. Thus, reflecting the abundance of highly repetitive DNA and dispersed repeats, which may be challenging to develop unique target sites for the majority of genes in maize [7].

Despite such challenges, CRISPR technology opens up the possibility for genome changes without foreign introgression of DNA vectors. CRISPR-Cas9 technology can be used as RNP complexes without the introgression and expression of a transgenic cassette in the host genome [8]. Such an approach would avoid a number of generations of backcrossing, expression vectors, and other invasive methods of cell penetration (e.g., biolistics) that can lead to gene disruption, including large deletions, partial trisomy, genome shattering events, and plant mosaicism [9]. Overall, these side effects can mask or interfere in the target gene functional analysis, and further additional biosafety concerns prior to commercial release.

CRISPR RNPs can be delivered directly to plant cells without the cell wall. Therefore, prior to transfection, the cell wall must be removed through enzymatic digestion reactions. Protoplast cells are viable in vivo biological material for DNA-free CRISPR delivery in plants. In addition, protoplasts are viable after transfection, which allows further tissue cultivation and propagation.

Delivery of preassembled Cas9 protein-gRNA RNPs or plant DNA-free genome-editing techniques are not exempt from off-target effects but represent an approach in which the effects of Cas9 could be isolated from other more invasive techniques [8,10,11]. This approach was first demonstrated in *Arabidopsis thaliana*, tobacco, lettuce, and rice protoplasts, including the regeneration of gene-edited lettuce [12]. After that, a few successful attempts were also accomplished on grapevine and apple [13], *Petunia × hybrida* [14], potato [15], and on soybeans and tobacco using CRISPR-Cpf1 (CRISPR from *Prevotella* and *Francisella*), recently named Cas12a (for review, please read Metje-Sprink et al. [8]). Maize and wheat plants with targeted mutations have also been successfully obtained by delivering gold particles coated with the RNPs into embryo cells (biolistics), followed by post-bombardment culture and plant regeneration [16]. However, the frequency of obtaining genome-edited plants was relatively low, since only 0.3–0.9% of regenerated maize plants possessed biallelic mutations [16].

The few studies published on maize genome editing rely mostly on the stable transformation [17,18,19,20]. In the manuscript, we delivered Cas9-gRNA RNP into maize leaf protoplasts via polyethylene glycol (PEG)-calcium mediated transfection, and indicated that In/Del mutations occurred with relatively high efficiency of 0.85–5.85% among the PEG-calcium-treated protoplast. We targeted the *inositol phosphate kinase* gene (IPK) involved in the phytic acid biosynthetic pathway. To develop a standard protocol for different maize varieties, we designed gRNAs and primers complementary to coding regions in exon three that are conserved in the species, in order to evaluate the efficiency and spectrum of DNA changes generated by CRISPR-Cas9 technology in maize, and also add relevant information to the safety of gene-edited organisms. This efficient and relatively easy assay method for the selection of gRNA suitable for editing of genes of interest will be highly useful for genome editing in maize, since the genome size and GC-content are large and high in the maize genome, respectively.

## 2. Materials and Methods

### 2.1. Target Site Selection and In Vitro Cleavage Assay

The *Zea mays* IPK gene sequence data was obtained from the NCBI GenBank (accession B73RefGen_v3). In vitro tests were performed to confirm the RNP complex efficiency to cleave target DNA. The target site was amplified using specific primers. The crispr-RNAs (crRNAs) were designed for the third exon of the IPK gene in maize using the platform CRISPR-Cas9 guide RNA Design Checker (Integrated DNA Technologies Inc. IDT, Coralville, IA, SUA). Commercially available Cas9 protein (160 kDa) was also purchased from IDT (Table 1).

gRNA, crRNA (100 nM), and trans-activating crispr-RNA (tracrRNA) (100 nM) were incubated for 5 min at 95 °C, according to the manufacturer’s instructions. Cas9 (100 nM), gRNA, and 1 × NEBuffer 3 (New England Biolabs Inc, NEB) were incubated for 10 min at 25 °C to form the RNP complex. Amplified PCR products (300 ng) were then incubated for 60 min at 37 °C with the RNP complex. Proteinase K (800U/mL) was added to stop the reaction. The products were visualized using 1% agarose gel electrophoresis [13].

### 2.2. Maize Protoplast Isolation and Fluorescent Transfection Assay

Mesophyll protoplasts were isolated following the protocol described by Sheen et al. [21] with some modifications. Etiolated maize seedlings were grown in vermiculite after disinfestation of the seeds with 70% alcohol (60 s), NaOCl2% (twice of 15 min), and triple washing with sterile water. Ten-days-old seedlings were used (three days under 16 h light/day and seven days in darkness). The middle portion of the second leaves were cut into thin strips (0.5–1 mm), and immersed in a cell-wall digestion enzyme solution (0.3% macerozyme R-10, 1.5% cellulase R-10, 10 mM of MES pH 5.7, 0.6 M mannitol, 10 mM CaCl_2_, 5 mM ß-mercapto, 0.1% BSA). The material was left in a vacuum for 30 min and gentle shaking at 40 rpm in the dark for 4 h. The protoplasts were released thoroughly by shaking at 80 rpm for 5 min After digestion, the protoplasts were diluted with the same volume of cold W5 solution [2 mM MES (pH 5.7), 154 mM NaCl, 125 mM CaCl_2_, 5 mM KCl] and filtered through a double filter (14 and 40 μM Nylon mesh). Protoplasts were collected after centrifugation at 100 g for 3 min and washed two times in 10 mL of W5 solution. Protoplasts were resuspended in cold MMG solution [0.4 M mannitol, 4 mM MES (pH 5.7), 15 mM MgCl_2_]. Its viability and concentration were determined using Fluorescein Diacetate (FDA; 0.2%) dye in the hemocytometer. To confirm the internalization of the RNP complex inside cells, an assay was performed using fluorescent-labeled tracrRNA molecules (ATTO 550, IDT) [9]. Microphotographs were taken using an inverted optic microscope ix80 Olympus.

### 2.3. Maize Protoplast Transformation

Maize protoplasts were gene-edited by introducing CRISPR-Cas9 RNP complex (no integration of exogenous DNA) via PEG-mediated transfection (Figure 1). Protoplast transformation was adapted from Woo et al. [12] and Malnoy et al. [13]. First, 15 μg of the two components of the gRNA (crRNA and trans-activating crispr RNA-tracrRNA)) were incubated at 95 °C for 5 min. After allowing them to cool at room temperature, 45 μg of Cas9 and 1 × NEBuffer 3 were added, then mixed, and incubated at 25 °C for 10 min. Finally, the RNP complex was mixed with 100 μL of protoplasts (1 × 10^5^ protoplasts), 250 μL of PEG solution (40% PEG 4000, 0.2 M mannitol, 0.1 M CaCl_2_) (pH 6.0), and incubated at 25 °C in the dark. Two incubation times were tested: T1 = 20 min and T2 = 40 min W5 solution (950 ul) was added and the tubes were centrifuged at 100 g for 3 min. Protoplasts were resuspended in 1 mL W1 solution [4 mM MES (pH 5.7), 0.5 M mannitol, 20 mM KCl], and then transferred to multi-well plates for 24 h under gentle agitation (40 rpm) in the dark at 25 °C.

### 2.4. Gene-Editing Efficiency Analysis by Sanger Sequencing

In order to characterize the spectrum and frequency of DNA changes at the target gene, genomic DNA was isolated using DNeasy^®^ Plant Mini Kit (QIAGEN^®^ Biotecnologia Brasil Ltd., Sao Paulo, SP, Brazil), followed by amplification of the target region by PCR using Taq Q5 High-Fidelity DNA Polymerase (NEB^®^, Ipswich, MA, USA) and primers listed in Table 1. PCR samples were purified and sequenced using the BigDye Terminator 3.1v Kit (ThermoFisher Scientific, Sao Paulo, SP, Brazil). Samples were resuspended in formamide, denatured at 95 °C for 5 min, and incubated on ice for 3 min. Sequencing was performed using the Sanger [22] automated sequencer from the 3500xL Dx Genetic Analyzer for Sequencing (Applied Biosystems^TM^-ThermoFisher Scientific, Sao Paulo, SP, Brazil).

CRISPR-Cas9 DNA changes were calculated based on the insertions and deletions (In/Dels) around the cleavage site (3 bp upstream of the PAM sequence) using the Inference of CRISPR Editing Software—ICE software v.2 (Synthego Corporation, Palo Alto, CA, USA). It has been previously shown that the ICE software results are comparable to the Next Generation Sequencing (NGS) results [23].

## 3. Results

### 3.1. In Vitro Cleavage Assay

Cleavage activity of gRNAs 1 to 5 was tested using 0.5 μg of crRNA and 1.5 μg of Cas9 enzyme to 300 ng of DNA. While all the designed gRNAs were able to cleave PCR products of the IPK gene in our study, the different gRNA sequences varied in their cleavage efficiency (Figure 2). gRNA2, 3, and 4 showed the highest activity and were, therefore, chosen for subsequent experiments on the transfection of maize protoplasts.

### 3.2. Targeted Mutagenesis in Maize Using CRISPR-Cas9 Ribonucleoproteins

Protoplasts viability was checked before and 1 h after transformation. The viability of protoplasts at the isolation step was on average 75% and on average more than 65% after transformation. Therefore, these protoplasts were considered stable after transformation, which would allow further cell culture propagation.

Frequently, results indicating low efficiency of CRISPR-Cas9 editing using RNPs delivery cannot discriminate low transfection rates from poor DNA cleavage and repair activity. In order to overcome this limitation and confirm the internalization of Cas9-gRNA RNPs, we performed a fluorescent microscopy assay. Labeled tracrRNA molecules confirmed the internalization of the RNP complex (Figure 3). Although this is not a quantitative method, it showed that at least one-third of the labeled molecules were internalized.

RNPs containing gRNAs 2, 3, and 4 were transfected into isolated protoplasts with PEG 4000 and the results are displayed as the percentage of indels detected at the cleavage site based on Sanger sequencing and analysis with the ICE software v.2 (Figure 4). The concentration of 45 ug of Cas9 and 15 ug of gRNA to 100 ul protoplasts in a 3:1 ratio resulted in the best cost-efficiency correlation. This result is also in agreement with previous reports on RNPs delivery into protoplasts, which ranged from 30–60 ug of Cas9 in a 1:1 and 3:1 ratio [12,13,24]. Exposure time was tested for all three gRNA sequences in 20- and 40-min of exposure. While a longer exposure time to the RNP complex led to a higher mutation index for all gRNAs tested, the increase in mutation rate was not consistent among all gRNA sequences (approx. seven-fold, three-fold, and one-fold for the gRNAs 2, 3, and 4, respectively). Deletions were shown more frequent than insertions in this model system. A higher insertion rate was only observed for gRNA 2 at 40 min time of exposure.

DNA sequences from the universal primer used to amplify the IPK gene in maize (876 bp) from treated samples were compared to the same fragment in negative controls (no RNP delivered). Other negative controls (Cas9 or gRNAs delivered alone) were also tested against the first negative control and showed no DNA changes. The obtained gene-edited sequences for each gRNA at 40 min of exposure are displayed in Figure 5. About six sequence variants were of major contribution in gene-editing efficiency for the three selected gRNA sequences.

Results on the percentage of mutated sequences (technology efficiency), the size of the DNA change (number of base pair change), the type of DNA change (deletions or insertions), and a theoretical knockout (KO) score are summarized in Table 2. The indel percentage at 20 min of exposure was, on average, 1.63% in contrast to 4.37% at 40 min-exposure time. Overall, gRNA 4 was most efficient and consistent at both exposure time. Intriguingly, gRNA 2 showed the lowest efficiency at 20 min (0.85%) but the highest efficiency at 40 min (5.85%). The Knockout Score accounts for the reads containing an amino acid frameshift change or 21 + bp indel. Thus, indicating the contributing indels that are likely to result in a functional knockout of the targeted gene. In the present study, the KO score was, on average, 0.83% for the 20 min and 3.17% for the 40 min of exposure treatment, which suggests that the majority of indels were frameshift modifications. In addition, only one gRNA at one time point presented a single base pair change as the most frequent mutation (gRNA 2 with a −1 bp). Notably, a deletion of 19 bp was the most abundant DNA change for gRNA 3 (0.9%). Moreover, all other gRNAs and exposure times showed a 2 bp deletion as the most frequent DNA change. Overall, the DNA change ranged from −28 bp to +12 bp change.

## 4. Discussion

CRISPR-Cas9 technology is a powerful tool for plant breeding and research. While still evolving as a technology to determine the rules for gRNA design and the algorithms to predict target and ‘off-target’ sequences, CRISPR applications still rely on empirical results to test the performance of new systems [25]. Notably, gene-editing results outcomes are frequently species-dependent [26]. Therefore, a CRISPR platform for different species with a rapid and efficient evaluation protocol is needed before commercialization.

Our experiments demonstrated the suitability of the PEG-delivered CRISPR-Cas9 RNPs system for gene-editing screening in maize. We showed that high-efficiency gene-edited maize cells could be obtained using less time-consuming (15 days) and labor-intensive procedures (PCR, agarose gel electrophoresis, and Sanger sequencing). In addition, the advantage of our system in relation to the use of vectors is that it prevents the integration and expression of exogenous DNA sequences, isolating the effect of gene-editing modification and avoiding transgene-introgressed side effects. Although non-integrating plasmids could be transfected into plant cells to deliver programmable nucleases, transfected plasmids are degraded in cells by endogenous nucleases, and the resulting small DNA fragments can be inserted at both on-target and off-target sites in host cells (Kim et al. 2014) [27]. For example, Braatz et al. [28] performed whole-genome sequencing after the transfection of the expression construct CRISPR-Cas9 in *Brassica napus* and found that the transformation resulted in at least five independent insertions of the vector backbone sequences in the plant genome.

### 4.1. Ribonucleoprotein Delivery in Plants

In the present report, we show a positive correlation between the time of exposure to RNPs and the efficiency of site-directed mutagenesis in maize, as ascertained with Sanger sequencing. In previous reports referring to the use of RNPs in plant protoplasts (Table 3), the authors used one or more different concentrations of RNPs and Cas9:gRNA ratios, but the effect of exposure time on mutation frequencies was not tested [29,30]. CRISPR RNPs were delivered to apple, grapevine, brassica sp., lettuce, tobacco, and rice plants at less or equal than 20 min exposure time and their efficiencies ranged between 0.1 and 40% [12,13,31]. In petunia and wheat protoplasts exposure for 30 min granted 0.2 up to 45% efficiency [14,17]; thus, suggesting that time of exposure might not alone explain indel frequency in different plant systems. In our system, when all other factors are maintained, exposure time consistently increased indel frequency for all three gRNA sequences tested (up to approx. seven-fold change increase).

The Cas9:gRNA ratio also influences target efficiency in a species-specific manner. Three different Cas9:gRNA ratios were tested in apple and grapevine protoplasts [13]. While the 1:1, 1:3, and 3:1 ratios did not differ in mutation frequency for grapevine (0.1%), the 1:1 and 3:1 ratios increased indel frequency in two (6.6 and 2.6 fold change respectively) out of three gRNA sequences in apple. Overall, the results obtained for the 3:1 ratio are equivalent to our results applying the same ratio (from 3.3% up to 6.7% efficiency). Cas9 concentration has been shown to be of major factor influencing the delivery of RNPs to plant cells. Woo et al. [12] tested 20 and 60 ug of Cas9 to *Arabidopsis* protoplasts and found that the editing efficiency was not directly related to the Cas9 concentration but also dependent on the time course of the analysis. At 24 h after delivery, more efficiency was observed when applying 20 instead of 60 ug of Cas9 (71 in contrast to 54%). Opposite results were obtained at 72 h after delivery. Nevertheless, increasing the amount of Cas9 (7.5, 15. 30, and 60 ug) was consistent with a crescent indel frequency in brassica sp. protoplasts [31]. The efficiency results obtained in our study were similar to those obtained applying approximately 60–90 ug of Cas9, thus, indicating that a lower amount of Cas9 (45 ug) but a higher exposure time (40 min) might have similar cleavage levels. Overall, it is clear that the limited amount of studies investigating the RNP delivery into plant cells is insufficient to draw definitive conclusions for increasing gene-editing efficiency using this system.

### 4.2. CRISPR Delivery Methods in Maize

Other delivery methods have been tested for maize as screening methods or gene-editing breeding methods, and these include PEG-mediated vector transfection, *Agrobacterium*-mediated, and biolistics (Table 3). The vast majority of studies still rely on vector-based transformation delivery of CRISPR. None of the listed studies have provided a cell-based screening method without the insertion of foreign vector-based DNA. On average, PEG-mediated vector transfection reached an average of about 12% efficiency, whereas *Agrobacterium* and biolistics reached 44% and 83%, respectively. In addition, most of these methods used embryogenic callus as explant material. Callus-based methods harbor chimeric tissues, thus, requiring subsequent genetic fixation to allow the stable inheritance of the edited traits. Therefore, these are not suitable material for genetic screening of successful gene-edited plants. Currently, many protocols are available for regeneration of whole plants from protoplasts. These include lettuce, tobacco, and rice, petunia, wheat, apple, and soybean. It is also suggested as a future choice for gene-edited maize and the list seems to expand because of the capabilities of the RNP technology [16,49]. The trade-off in efficiency percentage plays in return for avoiding unintended DNA integration and the potential undesirable biosafety risks.

### 4.3. Analytical Platforms for Gene-Editing Detection

Different analytical platforms for the detection and identification of CRISPR outcomes are reflected in frequency results as they show different analytical sensitivity. Woo et al. [12] showed a 40% transformation efficiency in tobacco when samples were analyzed by the Illumina sequencing platform. In contrast, the same samples showed a much lower efficiency rate (17%) when analyzed by the T7 cleavage assay. While T7 cleavage assays are nowadays being limited to a qualitative rather than quantitative detection methods, high-throughput sequencing platforms are time-consuming and costly options for screening protocols. Usually, such platforms are available in other labs or through service providers, which require long processing time and high costs for samples and assays that are still at the screening stage. In order to overcome such problems, we have proposed a model that analyzes gene-edited cell pools using data from common Sanger sequencing analysis. More specifically, we use the ICE: *Inference of CRISPR Edits* software, which enable the analysis of mixed populations and strongly correlates with next-generation sequencing of amplicons using Sanger sequencing data [23].

However, the proposed model is focused on providing a simple, cost-efficient analysis of gene-editing outcomes at the screening stage. The model is currently limited to detect mutations with more than 30 bp deletions or more than 14 bp additions. The analysis also does not account for very small mutant populations (<0.1%) neither presents mutations with a substitution of bp [23]. Although our platform does not apply exogenous DNA generating potential integration, it cannot be ruled out that microhomologies with gRNA sequences produce spurious cleavage or larges genomic rearrangements [50]. Therefore, long-run high throughput sequencing analysis is recommended for follow up breeding programs and safety tests [51].

## 5. Conclusions

We have shown that RNPs could be used for targeted CRISPR-Cas9 via PEG delivery as a model system to screen for gene-editing outcomes in maize. Target insertion and deletion DNA changes (In/Dels) were induced using 45 ug of Cas9 and gRNA used at a 3:1 ratio, and a positive correlation between exposure time (20 and 40 min) and indel frequency was observed. By targeting preserved coding regions, we can anticipate that the model could be applied to several maize varieties by validation using IPK gene. However, in vivo In/Del frequencies differed among gRNA sequences. In addition, the proposed method for sequencing analysis is also restricted to a window of 30 bp deletions and 14 bp additions. Further studies will be focused on CRISPR-Cas9 off-target activity and on the regeneration of edited protoplasts.

## Figures and Tables

**Figure 1 genes-11-01029-f001:**
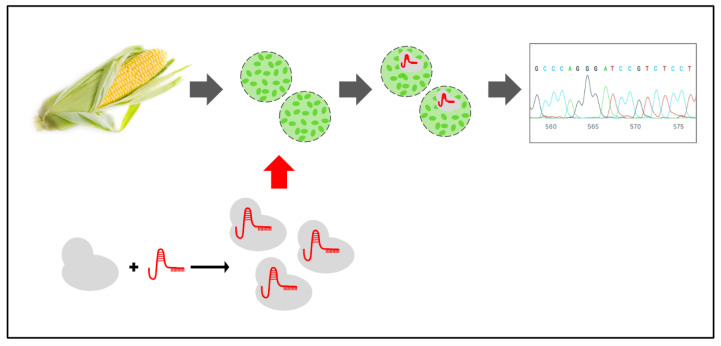
Methodological approach of the present study. Maize seeds were germinated *in vitro*; the second leaves of the seedlings were used to obtain the protoplasts. The protoplasts were exposed to the CRISPR-Cas9 ribonucleoproteins complex (RNP), and after 24 h, the DNA was extracted from the samples. PCR fragments were amplified and sequenced. Maize image was taken from the free repository from pexel.com.

**Figure 2 genes-11-01029-f002:**
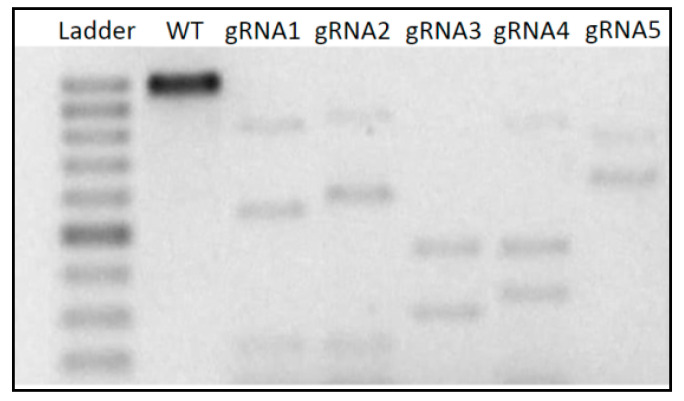
Schematic diagram of the Maize IPK gene locus with the guide RNAs (gRNAs) target sites. The in vitro CRISPR-Cas9 assay showing the original and the cleaved fragments of the IPK gene in maize that were submitted to the RNP complex with the crRNA1, 2, 3, 4, and 5. Note: WT = Wild Type (control).

**Figure 3 genes-11-01029-f003:**
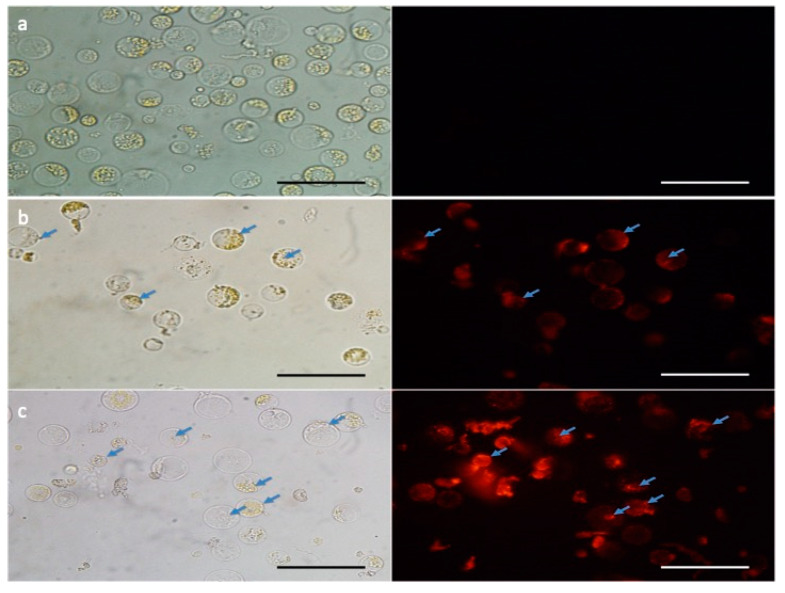
Microscope photographs of maize protoplast cells transfected with polyethylene glycol (PEG)-mediated CRSIPR-Cas9 RNP. The RNP containing the ATTO550-conjugated trans-activating crispr RNA (tracrRNA) was transfected into maize protoplasts and cells were monitored under fluorescent microscopy to check the transfection efficiency. Comparing to the cells without CRISPR-Cas9 delivery (**a**—control sample), the red fluorescent signal was detected in CRISPR-Cas9 treated samples (**b** and **c**—right). Photographs were taken with white light (left side) and fluorescent light (right side). Blue arrows indicate the internalization of the RNP complex. Scale bars: 100 μm.

**Figure 4 genes-11-01029-f004:**
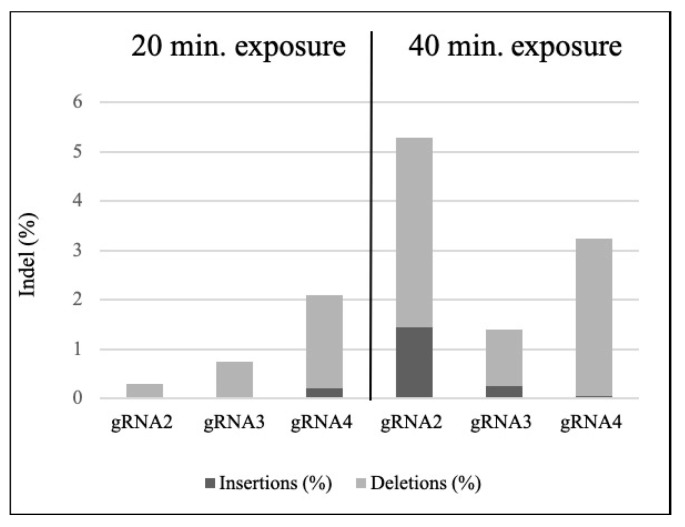
Frequency of mutations in maize protoplasts generated by CRISPR-Cas9 and measured by the Inference of CRISPR Editing Software—ICE software v.2. Different gRNAs and exposure time of the protoplasts to the RNP complex are represented. Percentages of insertions and deletions are represented in dark grey and light grey, respectively.

**Figure 5 genes-11-01029-f005:**
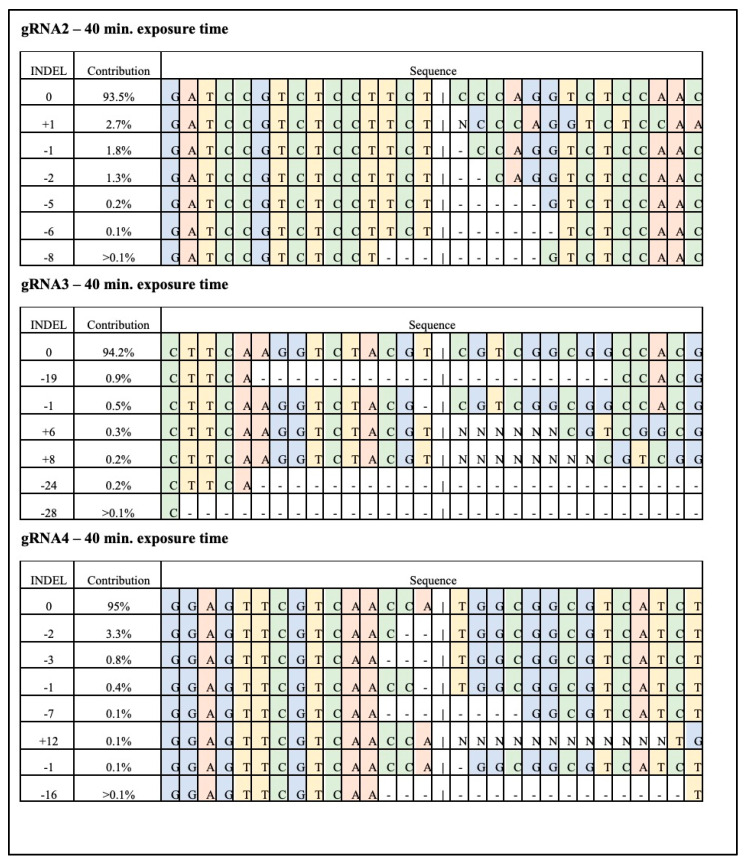
Sequence distribution of most efficient mutations identified with the ICE software around the IPK gene target site in *Zea mays*. Edited sequences were obtained after CRISPR-Cas9 RNP transfection to maize protoplasts. Forty-five μg of Cas9 preassembled with 15 μg of each gRNA were used in the protoplast transformation. Different exposure times of the RNP complex with the protoplast are presented. Cut sites are represented by black vertical dotted lines, insertions are represented by ‘N’ and deletions by black horizontal dotted lines.

**Table 1 genes-11-01029-t001:** List of primers and crispr-RNAs (crRNAs) used for the amplification and mutation of the *inositol phosphate kinase* gene (IPK) target locus in maize via CRISPR (Clustered Regularly Interspaced Short Palindromic Repeats)-Cas9.

**Primer**	**Sequence (5′–3′)**	**Amplicon Size (bp)**
ZmIPK-F	GAAGAAGCAGCAGAGCTTCA	876
ZmIPK-R	CAGAAGAAATCCGTGAGGACAG	
**crRNA**	**Sequence (5′–3′)**	**Cleavage Fragments (bp)**
crRNA1	AGCTCGACCACGCCGCCGAC	279 | 597
crRNA2 *	GGGATCCGTCTCCTTCTCCC	617 | 259
crRNA3 *	ATCTTCAAGGTCTACGTCGT	525 | 351
crRNA4 *	CAGGAGTTCGTCAACCATGG	498 |378
crRNA5	ACAAGCTCTACGGAGACGAC	141 | 735

Note: * Selected crRNAs used for protoplasts transfection.

**Table 2 genes-11-01029-t002:** Mutation rates in *Zea mays* IPK gene target region produced by CRISPR-Cas RNP delivery and analyzed by Sanger sequencing and ICE software analysis.

Sample	Incubation Time (min)	% of Indel	Model Fit (R^2^)	KO Score	Mutation Range (bp)	Greater Contribution (bp)
Cas9 only	20	0		0	0	0
gRNA2 only	20	0		0	0	0
gRNA3 only	20	0		0	0	0
gRNA4 only	20	0		0	0	0
Cas9 + gRNA2 rep1	20	0	0.99	0	0	0
Cas9 + gRNA2 rep2	20	1	1	1	−4 to −2	−2
Cas9 + gRNA3 rep1	20	0	1	0	−7 to −1	−2
Cas9 + gRNA3 rep2	20	1	1	1	−7 to−1	−2
Cas9 + gRNA4 rep1	20	3	0.99	2	−7 to +12	−2
Cas9 + gRNA4 rep2	20	1	0.99	1	−2 to +3	−2
Cas9 + gRNA2 rep1	40	4	0.99	4	−7 to +1	−1
Cas9 + gRNA2 rep2	40	6	1	6	-8 to +1	+1
Cas9 + gRNA3 rep1	40	1	1	1	−19 to−1	−1
Cas9 + gRNA3 rep2	40	2	0.96	2	−28 to +8	−19
Cas9 + gRNA4 rep1	40	2	0.99	2	−3 to −2	−2
Cas9 + gRNA4 rep2	40	5	1	4	−16 to +12	−2

Note: KO = Knockout.

**Table 3 genes-11-01029-t003:** Publications with DNA-free gene editing in plants using CRISPR-Cas9 RNPs and other delivery methods for maize.

Reference	Plant Species	Plant Material	Transfection Method	Gene-Editing Efficiency
RNP delivered in plants				
[12]	*Arabidopsis thaliana*, *Lactuca sativa*, *Nicotiana attenuata*, *Oryza sativa*	Protoplasts	PEG-mediated	5.7–40.0%
[13]	*Malus domestica, Vitis vinifera*	Protoplasts	PEG-mediated	0.1–6.9%
[14]	*Petunia x hybrid*	Protoplasts	PEG-mediated	2.4–21.0%
[31]	*Triticum aestivum*	Protoplasts, immature embryos	PEG-mediated, Biolistics	0.2–45.3%
[15]	*Solanum tuberosum*	Protoplasts	PEG-mediated	1.0–25.0%
[32]	*Brassica oleracea, Brassica rapa*	Protoplasts	PEG-mediated	0.1–24.5%
[33]	*Oryza sativa*	Zygotes	PEG-mediated	14.0–64.0%
Other delivery methods in maize				
[34]	*Zea mays*	Protoplasts	Vector via PEG-mediated	13.1%
		Immature embryos	*Agrobacterium*-mediated	16.4–19.1%
[35]	*Zea mays*	Protoplasts	Vector via PEG-mediated	N.A
[24]	*Zea mays*	Immature embryos	Vector via biolistics	1.3–4.6%
[17]	*Zea mays*	Protoplasts	Vector via biolistics	80.0–90.0%
[36]	*Zea mays*	Immature embryos	*Agrobacterium*-mediated	57.1–71.4%
[19]	*Zea mays*	Protoplasts	Vector via PEG-mediated	2.8–27.0%
		Immature embryos	*Agrobacterium*-mediated	19.0–31.0%
[37]	*Zea mays*	Protoplasts	Vector via PEG-mediated	4.0–11.9%
		Immature embryos	*Agrobacterium*-mediated	65.8–86.9%
[16]	*Zea mays*	Immature embryos	RNP via Biolistics	0.01–0.7%
[18]	*Zea mays*	Immature embryos	*Agrobacterium*-mediated	12.0–74.0%
[38]	*Zea mays*	Immature embryos	Vector via Biolistics	60.0–98.0%
[39]	*Zea mays*	Immature embryos	*Agrobacterium*-mediated	N.A
[19]	*Zea mays*	Immature embryos	*Agrobacterium*-mediated	5.0–100%
[20]	*Zea mays*	Immature embryos	*Agrobacterium*-mediated	N.A
[40]	*Zea mays*	Immature embryos	*Agrobacterium*-mediated	90.0–100%
[41]	*Zea mays*	Immature embryos	*Agrobacterium*-mediated	N.A
[42]	*Zea mays*	Immature embryos	*Agrobacterium*-mediated	N.A
[43]	*Zea mays*	Immature embryos	*Agrobacterium*-mediated	N.A
[44]	*Zea mays*	Immature embryos	*Agrobacterium*-mediated	N.A
[45]	*Zea mays*	Immature embryos	*Agrobacterium*-mediated	N.A
[46]	*Zea mays*	Immature embryos	Vector via biolistics	N.A
[47]	*Zea mays*	Immature embryos	*Agrobacterium*-mediated	25.0–100%
[48]	*Zea mays*	Immature embryos	*Agrobacterium*-mediated	N.A
Present study	*Zea mays*	Protoplasts	RNA via PEG-mediated	0.85–5.85%

Note: N.A is ‘not applicable’ because transgenic plants were either selected using antibiotic marker genes or the analysis was not performed.

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
