# Peer review of "PEG-Delivered CRISPR-Cas9 Ribonucleoproteins System for Gene-Editing Screening of Maize Protoplasts"

_genes, 2020, doi:10.3390/genes11091029_

Round 1

Reviewer 1 Report

In this manuscript author explan about PEG-delivered CRISPR-Cas9 Ribonucleoproteins system for gene-editing screening of maize protoplasts. Author claimed to have performed the transformation of maize protoplasts using different gRNAs sequences targeting the inositol phosphate kinase gene and applying two different exposure times to RNPs. The overall manuscript is Ok but written poorly and has many grammatical mistakes. I also found this MS is plagiarised at many places.

Major:

  1. In figure 3 add bar and add one more lane of the merged picture.
  2. How much stable protoplasts were?
  3. Check the knockout expression of inositol phosphate kinase gene and IP6 content in KO line compared with WT.

Minor:

L45 cell natural to natural cell.

L48 unprecedent  to unprecedented.

L69 Such approach to Such an approach.

L119 deedlings to seedlings.

L130 fluorescent labelled to fluorescently labeled.

L178 one third to one-third.

L179 labelled to labeled.

L200, L208, L210, L212 40 min exposure to 40 min. of exposure.

L222 to a 4.37% to to 4.37%.

L223 efficient to efficiency.

L239 relies on to rely on.

L241 for difference species to for different species.

L277 has shown to has been shown.

L280 dependent to the time course to dependent on the time course.

L284 study was similar to study were similar.

L306 outcomes reflects to outcomes reflect.

L312 costly option for to costly options for.

L313 requires a long processing time to requires long processing time.

L316 which unable the analysis of mixed to which enable the analysis of mixed.

L322 present mutations to presents mutations.

L325 Therefore, long run high through put sequencing analysis are to Therefore, long-run high throughput sequencing analysis is.

L19-L20, L28, L45-47, L81-82, L257-258, L228  have been plagiarised from https://doi.org/10.3389/fpls.2018.01594 this frontiers paper.

L42-43, L192, L226, L282, also been plagiarized.

Author Response

Reply to reviewer #1 in attached file.

Reviewer 2 Report

This manuscript shows interesting data. Nevertheless, there are some structural issues that should be addressed in order to take a properly informed and adequate decision.

Introduction

I suggest reviewing, the organization and structure of the introduction.

It would be nice if the authors provide information regarding:

  • short introduction of the origin and domestication of maize,
  • gene editing and CRISPR Cas technique
  • application of biotechnologies , including gene editing, in maize
  • protoplast tool (I suggest to dispalce figure 1 in this part)

Results

Please provide to add the result for protoplast isolation. Moreover please add  a new chapter for the detection of mutation.

In figure 5 please add: Total Indel%’ (the editing efficiency, or the percent of the total pool of protoplast samples that are not wild-typesequences) and ‘Knockout Score’ ( the proportion of protoplasts in each sample that either have a frame shift mutation or have an indel of 21 nucleotides or more).

Line 38 Please change "emergence"

Line 44 Please change "guiding" with "to lead"

Line 181 Please add "a" and "b" in figure 3

Author Response

Reply to reviewer #1 in attached file.

Round 2

Reviewer 1 Report

I am happy with the responses of authors